# Vaccination against Community-Acquired Pneumonia in Spanish Adults: Practical Recommendations by the NeumoExperts Prevention Group

**DOI:** 10.3390/antibiotics12010138

**Published:** 2023-01-10

**Authors:** Esther Redondo, Irene Rivero-Calle, Enrique Mascarós, Daniel Ocaña, Isabel Jimeno, Ángel Gil, José Luis Díaz-Maroto, Manuel Linares, María Ángeles Onieva-García, Fernando González-Romo, José Yuste, Federico Martinón-Torres

**Affiliations:** 1Infectious, Migrant, Vaccines and Preventive Activities Group of SEMERGEN, International Healthcare Center of Ayuntamiento de Madrid, 28006 Madrid, Spain; 2Translational Pediatrics and Infectious Diseases Section, Pediatrics Department, Hospital Clínico Universitario de Santiago de Compostela, 15706 Santiago de Compostela, Spain; 3Vaccines, Infections and Pediatrics Research Group (GENVIP), Healthcare Research Institute of Santiago de Compostela, 15706 Santiago de Compostela, Spain; 4CIBER of Respiratory Diseases (CIBERES), Instituto de Salud Carlos III, 28029 Madrid, Spain; 5Health Department, Hospital la Fe, Primary Care Center Arquitecto Tolsá, 46019 Valencia, Spain; 6Primary Care Health Care Center Algeciras, 11204 Algeciras, Spain; 7Primary Care Health Center Isla de Oza, Vaccine Responsible of SEMG, 28034 Madrid, Spain; 8Preventive and Public Health Department, Rey Juan Carlos University, 28922 Madrid, Spain; 9Primary Care Health Center Guadalajara, Infectious Diseases Group SEMERGEN, 19161 Guadalajara, Spain; 10Infectious Diseases Group SEMERGEN, Fundación iO, 28006 Madrid, Spain; 11Preventive Medicine Department, Hospital Costa del Sol, 29603 Marbella, Spain; 12Clinical Microbiology Department, Health Research Institute (IdISSC), Hospital Clínico San Carlos, School of Medicine, Complutense University of Madrid (UCM), 28040 Madrid, Spain; 13National Center for Microbiology, Instituto de Salud Carlos III, 28220 Madrid, Spain

**Keywords:** community-acquired pneumonia, influenza vaccination, pneumococcal conjugate vaccines, pneumococcal polysaccharide vaccine, SARS-CoV-2 vaccines, COVID-19, pertussis, influenza vaccines, VRS, antimicrobial resistance

## Abstract

In the adult population, community-acquired pneumonia (CAP) is a serious disease that is responsible for high morbidity and mortality rates, being frequently associated with multidrug resistant pathogens. The aim of this review is to update a practical immunization prevention guideline for CAP in Spain caused by prevalent respiratory pathogens, based on the available scientific evidence through extensive bibliographic review and expert opinion. The emergence of COVID-19 as an additional etiological cause of CAP, together with the rapid changes in the availability of vaccines and recommendations against SARS-CoV-2, justifies the need for an update. In addition, new conjugate vaccines of broader spectrum against pneumococcus, existing vaccines targeting influenza and pertussis or upcoming vaccines against respiratory syncytial virus (RSV) will be very useful prophylactic tools to diminish the burden of CAP and all of its derived complications. In this manuscript, we provide practical recommendations for adult vaccination against the pathogens mentioned above, including their contribution against antibiotic resistance. This guide is intended for the individual perspective of protection and not for vaccination policies, as we do not pretend to interfere with the official recommendations of any country. The use of vaccines is a realistic approach to fight these infections and ameliorate the impact of antimicrobial resistance. All of the recently available scientific evidence included in this review gives support to the indications established in this practical guide to reinforce the dissemination and implementation of these recommendations in routine clinical practice.

## 1. Introduction

Before the recent pandemic caused by SARS-CoV-2, infectious diseases were responsible for one third of the annual deaths worldwide, and indeed, infections affecting the lower respiratory tract were responsible for 4 million deaths globally [1]. According to the World Health Organization, pneumonia kills more children worldwide than any other disease, even more than acquired immune deficiency syndrome (AIDS), malaria and measles combined [2,3,4]. In the adult population, this is even worse because the impact of community-acquired pneumonia (CAP) or nosocomial pneumonia (including hospital-acquired pneumonia and ventilator-acquired pneumonia) is also remarkably alarming, as they are associated with high morbidity and mortality rates worldwide [5]. The pandemic caused by SARS-CoV-2 has dramatically exacerbated this scenario, as COVID-19 deaths were estimated at 18.2 million worldwide within the first two pandemic years [6]. From the antimicrobial resistance perspective, the COVID-19 pandemic has also contributed to the rise in resistant strains, probably caused by the higher empirical use of antibiotics to avoid co-infections with bacterial pathogens in patients infected with SARS-CoV-2 [7,8]. In Spain, similar findings have been described as showing an increased proportion of pneumococcal strains with reduced susceptibility to several antibiotics [9]. In this sense, implementation of effective vaccines with high coverage rates for both the pediatric and adult population is the best strategy to decrease the burden of disease caused by multidrug resistant (MDR) pathogens and avoid the spread of resistance genes [10,11]. Current vaccines against relevant bacterial pathogens have contributed to diminish the rates of infection caused by strains harboring antimicrobial resistance [12,13,14,15]. Hence, two of the best examples are the significant reduction in serotypes associated with MDR after the introduction of pneumococcal conjugate vaccines (PCVs) [9,16] and the decrease in ampicillin resistance strains after the use of the vaccine against *Haemophilus influenzae* type B [17].

In addition, influenza virus vaccines may reduce subsequent infections by some of the bacterial pathogens that largely co-infect the respiratory tract, producing severe pneumonia [10,18]. Hence, co-infections of influenza virus and *Streptococcus pneumoniae* are well known [19,20], although many other bacterial pathogens, including *Staphylococcus aureus*, *Neisseria meningitidis*, *Streptococcus pyogenes* and *Haemophilus influenzae*, can induce severe infections in patients co-infected with influenza virus [21,22,23]. The estimated prevalence of atypical pathogens in hospitalized CAP patients is low during a non-epidemic season [24]. The use of COVID-19 vaccines could also reduce the severity of the pneumonia process caused by MDR strains because, in many cases, co-infection with COVID-19 requires ICU admission [25,26].

In Spain, CAP constitutes the first cause of mortality by an infectious disease [27]. Based on the lack of specific recommendations to prevent the appearance of CAP, the NeumoExperts Prevention group (NEP) initially developed a practical vaccination guide to prevent CAP in the year 2016; this guide has been widely distributed and adopted by health professionals within the country [28]. This guide was established using the available scientific literature in order to facilitate the clinical practice of all first-line health professionals involved in the vaccine and immunization field. These recommendations were updated in 2019 and 2021 [29,30]; however, due to the commercialization of newer vaccines for *S. pneumoniae* based on PCVs of broader coverage and the introduction of new or adapted COVID-19 vaccines, a revised version is necessary. In addition, we believe that this updated guideline may be useful and eventually extrapolated to other countries (Figure 1 and Figure 2).

## 2. Materials and Methods

We applied the same methodology described in previous guideline editions [28,29,30]. It consisted of an extensive medical literature search using Medline and Cochrane databases including national and international recommendations targeting prevention of CAP in adults. During this process, the authors of this manuscript had at least two virtual and two face-to-face meetings to discuss the positioning of our group in this field. All of the vaccine recommendations against CAP in adults were discussed and approved based on the best available evidence following the expert opinions described in the Oxford criteria [31].

## 3. Results

### 3.1. Vaccination against Streptococcus pneumoniae

Prevention of pneumococcal disease in adults can be achieved by using PCVs, the 23- valent polysaccharide vaccine (PPSV23) or the sequential schedule of PCVs and the PPSV23. The serotypes covered by the different pneumococcal vaccines that are currently licensed in Europe are indicated in Figure 3.

Based on the heterogeneous policy of vaccination in Spain, with 19 different regions displaying specific vaccination policies, the use of PCVs in the pediatric population has been heterogeneous among the different Spanish regions. PCV7 was firstly commercialized in 2001, although its use was mainly based in the private market. PCV10 was authorized in 2009, but it was rapidly replaced by PCV13, which become available in 2010 for the private market. In 2015, the Spanish Ministry of Health approved the systematic use of PCV13 for the pediatric population, following a 2 + 1 schedule. In adults, the vaccination strategy is even more heterogeneous, affecting the entire country. For immunocompetent adults aged ≥65 years, the systematic use of PPSV23 has been the general recommendation by the Ministry of Health since 2004, although different regions have included the use of PCV13 for adults. In high-risk patients, the national recommendation is the use of the sequential strategy (PCV13 followed by PPSV23).

In terms of sequential strategies, compliance is an important aspect to be considered in the vaccination schedules in order to obtain the maximum benefit with the different vaccines. In this sense, recent studies have evaluated the situation in the USA, citing a low completion rate in sequential schedules [32]. In a second study evaluating compliance with the recent ACIP pneumococcal vaccination recommendations, a low completion rate was also observed in the age group of adults > 65 years old and in the cohorts of high-risk patients and those with underlying conditions [33]. Overall, both studies described above are critics of the sequential schedules due to low compliance; therefore, we suggest that a single dose with a broader PCV would increase the coverage in comparison to sequential strategies.

#### 3.1.1. Burden of Pneumococcal Disease in Spain

The evolution of IPD in the Spanish pediatric population has demonstrated a clear impact from vaccination with PCV13 during the last decade (2009–2019). In children aged <2 years, the incidence of PCV13 serotypes declined from almost 27 cases per 100,000 population in the pre-PCV13 period (2009) to around 4 cases in the last period (2018–2019), demonstrating a 90% reduction in IPD cases when comparing 2009 vs. 2019 [34]. In children aged 2–5 years, there was an 88% reduction in IPD cases when comparing 2009 and 2019, confirming that vaccination with PCV13 in Spanish children progressively reduced the incidence of IPD, although a rise in non-PCV13 serotypes was also observed in the last years [34]. In adults, the burden of disease by all serotypes remained unchanged when the pre-vaccine period (year 2009) and the last period studied (2019) were compared. However, when PCV13 serotypes were analyzed, there was a reduction of up to 67% in adults aged 18–64 years and 50% in adults aged ≥65 years when the pre-vaccine period (2009) and the year 2019 were compared [34]. These data confirm that vaccination with PCV13 in Spanish children was a very effective measure that significantly reduced IPD episodes caused by vaccine-preventable serotypes in the adult population, demonstrating an important herd protection effect for Spanish adults. However, specific and unique serotypes contained in PPSV23 but not in PCV13 had largely increased since 2014, leading to incidence rates in 2019 very similar to those in the pre-vaccine period (2009). These results confirmed that the use of PPSV23 in Spain did not have any global impact on the burden of disease in the adult population [34]. The rise in unique serotypes covered by PPSV23 was mainly due to the emergence of serotype 8, followed by serotype 22F in adults of all ages. Hence, serotypes 8 and 3 accounted for up to 32% of all IPD cases in the adult population aged ≥65 years [34].

In terms of CAP, a recent multicenter study including different Spanish hospitals has shown a similar impact of PCV13 in adults. Serotypes covered by PCV13 that cause CAP showed a substantial reduction in immunocompetent adults after the introduction of pediatric vaccination with PCV13, showing again a positive herd effect against CAP in adults [35]. Serotypes 3 and 8 were the most prevalent serotypes causing CAP during the recent years of the study (2016–2018). In addition, serotype 3 was found in almost 25% of pneumococcal CAP associated with complications, being very prevalent in patients with renal failure, pleural effusion and those requiring invasive mechanical ventilation [35].

Another important aspect to be considered from the epidemiological perspective is the impact of vaccine programs against MDR strains. In this sense, the use of PCV7 followed by PCV13 in Spanish children reduced the incidence of vaccine serotypes associated with antibiotic resistance in adults. However, an increase in non-PCV13 serotypes displaying resistance to β-lactams, macrolides and even MDR has occurred in the last few years in the adult population in Spain [16]. Serotypes 11A and 24F were the most prevalent non-PCV13 serotypes with reduced susceptibility to penicillin and other β-lactams in Spanish adults [9]. In the case of serotype 11A, this is of great importance because during the pre-pandemic period (2016–2019), the MIC_90_ of penicillin against this serotype was 2 µg/mL, which is considered susceptible with increased exposure following EUCAST breakpoints. However, during the first pandemic year (2020), the MIC_90_ against serotype 11A in Spain increased to 4 µg/mL, which is the breakpoint considered as resistant [9]. The generic use of antibiotics during 2020 in patients infected with SARS-CoV-2 increased the proportion of pneumococcal strains in adults in Spain resistant to different β-lactams and macrolides and might be the reason for the increased MIC_90_ levels against serotype 11A [9]. This is of great relevance because serotype 11A is one of the most lethal serotypes in Spain (even higher than the mortality caused by serotype 3) probably due to the antimicrobial resistance pattern [36] but also because it is a serotype with a marked ability to avoid complement-mediated phagocytosis and induce biofilms [37].

#### 3.1.2. Position of NEP Group on Pneumococcal Vaccination in Adults

Since the first recommendation proposal to prevent pneumonia, the NEP group has promoted the use of PCVs for adults as a major point of the vaccine calendar. This recommendation is justified by the type and duration of the immune response, the effectiveness against mucosal disease, the overall reduction of the disease burden and the ability to control the spread of serotypes associated with antimicrobial resistance.

In recent years, new scientific evidence addressing pneumococcal vaccination has been published and should be considered in order to establish new recommendations. The most important achievements are described as follows:
1.PCV13 has been demonstrated to be effective against pneumonia, IPD and antibiotic-resistant strains in adults.2.The use of pediatric vaccination with PCVs is not enough to eliminate the burden of disease in adults, especially against mucosal infections, despite the indirect effects of these vaccines.3.The polysaccharide vaccine (PPSV23), after being routinely used for many years, has not controlled the epidemiology of specific and unique serotypes contained in this vaccine, probably due to the different immune response elicited in comparison to PCVs.4.Previous administration of polysaccharide vaccines can interfere with the immune response of subsequent conjugate vaccines.5.Sequential vaccination strategies have the disadvantage of poor compliance by the targeted population. This approach also increases the attendance rates at primary care centers and hospitals for the next doses, including more human resources costs and the possibility of inducing mistakes by health care professionals regarding the correct sequential posology and dose intervals.6.The commercialization of new PCVs of broader spectrum (PCV15 and PCV20) and the development of new vaccines (PCV21, PCV24 and PCV26) confirm our initial recommendation of using PCVs instead of PPSV23 for adults and reinforce the natural strategy of replacing plain polysaccharide vaccines with PCVs with greater coverage, as has happened with other infections such as meningococcal disease.7.The new generation of PCVs (PCV15 and PCV20) has been approved exclusively on the basis of non-inferiority immunological criteria, and there are no studies directly comparing these new PCVs.8.Policy decisions regarding the ideal vaccination strategy should include the following considerations:Epidemiological context at national level and not only using extrapolated data from other populations and countries with different burden of disease, circulating serotypes and vaccine calendars;The potential coverage of the different PCVs and their potential impact against the major forms of pneumococcal disease in our area;The relevance of specific serotypes based on their prevalence, clinical phenotype and/or antibiotic resistance in our geographic area (i.e., serotypes 3, 8, 11A or 22F);The lack of studies directly comparing the new PCVs and evaluating the clinical impact of the immunogenic differences between these vaccines against shared serotypes. These data are difficult to interpret in terms of clinical effectiveness, especially the prevention of mucosal disease that is, indeed, the most prevalent in adults;The degree of compliance in the sequential schedule;The intervention efficiency.

Based on all of the considerations described above, the NEP group recommends that PCVs should play a key role in the prevention strategy against pneumococcal pneumonia and IPD, indicating two possible practical schedules for vaccination in adults:Vaccination with a single dose of PCV20Sequential schedule using PCV15 followed by PPSV23

However, considering that the use of a single dose of PCV20 would cover the majority of current pneumococcal diseases in Spain, the first option using a single dose of PCV20 would facilitate compliance with the vaccine strategy, and it would avoid interference with future PCVs. With this recommendation, we assume that a limited proportion of serotypes present in PPSV23 and not in PCV20 would not be eventually covered by our proposal (up to 3% in pneumococcal pneumonia and 8% in IPD) [34,35].

In addition, reflecting the NEP group’s position, we recommend vaccination against *S. pneumoniae* for all adults ≥ 60 years old and also patients belonging to risk groups, regardless of the vaccine strategy used. It is essential to analyze specific scenarios to determine the need for booster doses and the characteristics and intervals of these additional doses in individuals previously vaccinated with PCV13 and/or PPSV23 [38]. Finally, a common national immunization calendar against pneumococcal pneumonia for Spanish adults across the different regions would be recommendable, and the proposed schedule of age-based vaccination with a single dose of PCV20 might facilitate this task.

### 3.2. Vaccination against Influenza

#### 3.2.1. Burden of Disease and Vaccine Coverage in Spain

The influenza virus showed a dramatic decrease in terms of disease incidence from the national and global perspectives from the start of the COVID-19 pandemic until May 2022 [39,40,41]. Non-pharmacological interventions, such as face masks, lockdowns, social distancing, and even viral interference with the SARS-CoV-2 virus, have played a fundamental role in the scarce incidence that has been recognized by the World Health Organization (WHO) [42].

During 2022, in Spain, we found a variation in the seasonal pattern for influenza, with a moderate recovery in the number of cases in atypical dates covering the period from May to August, with influenza A (H3N2) as the predominant strain that was stable until the beginning of the typical flu season [41]. The current situation in the southern hemisphere [43], with persistent episodes, might be a pattern that can be found during the current 2022–2023 season in the northern hemisphere, including Spain.

In the previous season (2021–2022), the vaccine coverage reported by the Spanish Ministry of Health was higher than that obtained in the pre-pandemic period, but only in the age group of 75 years old. These rates reached 75% coverage, which is the vaccine coverage recommended by WHO for older patients and those with comorbidities [44]. In Spain, the coverage for the age group of ≥65 years old was 69.4%, confirming an increase in comparison to previous seasons. In addition, the vaccine coverage in pregnancy was lower in comparison to the previous season (54.98% in 2021–2022 vs. 62.3% in the 2020–2021 season). In the case of healthcare workers, the average vaccine coverage was 60%, although it was variable within the different Spanish regions, with Valencia region being at the top with vaccine rates of up to 95% [45].

The vaccine coverages in the age group ≥ 65 years have been in agreement with the results published recently in the “Gripómetro”, a demographic study designed to monitor, in real time, national and regional influenza vaccine coverage on a weekly basis in Spain [46]. This tool was evaluated by a group of experts, concluding that the data obtained were closely correlated with the national data supplied by the Spanish Ministry of Health at the end of every flu season [46].

It will be essential to know the vaccine coverage in the risk groups with comorbidities and not only publish the vaccine coverage for the elderly population and pregnant women. A recent study reporting on the evolution of influenza including up to 10 different seasons concluded that 18.9% of hospitalizations for flu in Spain were caused in patients < 65 years old with comorbidities, but this rate increased up to 40.1 % in people ≥ 65 years old with underlying diseases [47]. In fact, an analysis evaluating direct costs at the national level for vaccine-preventable infectious diseases confirmed that 51.5% of all direct costs attributed to the influenza virus occurred in the age group ≥45 years old. These costs were calculated at almost EUR 56 million and were mainly associated with hospitalizations (43.1%) and primary care centers (35.4%) [48].

The main goal of vaccination programs against influenza would be the reduction of flu infections and also the complications attributed to the disease, mainly from the systemic perspective, especially in patients with comorbidities. In this context, the results of a randomized clinical trial including 2532 participants were recently published, evaluating flu vaccination for the prevention of cardiovascular complications in patients with cardiovascular risk. Flu vaccination demonstrated a reduction in mortality from all causes and a decrease in the risk of suffering myocardial infarct and/or stent-thrombosis in 28% of patients (hazard ratio, 0.72 [95% CI, 0.52–0.99]; *p* = 0.040). This reduction increased up to 41% when mortality was associated with any cardiovascular event or mortality from all causes (hazard ratio, 0.59 [95% CI, 0.39–0.89]; *p* = 0.010 and hazard ratio, 0.59 [95% CI, 0.39–0.90]; *p* = 0.014, respectively) [49]. This benefit was also observed in a recent meta-analysis of randomized clinical trials evaluating the impact of flu vaccination on cardiovascular risk, confirming that flu vaccination was associated with a 34% reduction in the risk of suffering a severe cardiovascular event [50].

In addition, different studies have demonstrated that flu vaccination prevents cerebrovascular accidents in Spain [51] and Taiwan [52], and in both cases, flu vaccine reduces the risk of these pathologies. Another study, including almost 23 million hospitalized patients, demonstrated that cardiovascular complications and cerebrovascular accidents were significantly reduced after flu vaccination alone or in combination with anti-pneumococcal vaccine [53].

In diabetic patients, a Spanish study demonstrated that those who received flu vaccination had a 46% lower risk of hospitalization [54]. The reduction of complications associated with influenza infection is critical, because patients with type 2 diabetes had up to 75% risk of abnormal increased glycemic events when suffering a flu infection [55].

#### 3.2.2. Recommendations for Influenza Vaccination

During the current 2022–2023 epidemiologic season, all of the flu vaccines used in the different Spanish regions will be tetravalent. In addition, during this season, three Spanish regions (Galicia, Murcia and Andalusia) will include the universal vaccination of children between 6–59 months of age, following the recommendations of the Spanish Pediatric Association [56]. As a novelty for this year, an attenuated flu vaccine for intranasal inoculation will be available at pharmacies across the entire country, with the indication for children between 2–17 years of age [57]. In addition, the Spanish Ministry of Health has included universal pediatric vaccination for this age group in its recommendation for the next season, 2023–2024 [58]. For the current season, 2022–2023, there are some novelties regarding flu vaccination affecting the elderly and/or nursing homes and even highly vulnerable patients using the tetravalent recombinant flu vaccine. It will be the first time that this type of immunization is included in a vaccination program in Spain. Moreover, the high-dose tetravalent vaccine, which in the previous season was the most widely used vaccine in Spain for the elderly population living in nursing homes, has been selected as the first choice for the general population ≥ 80 years old and also for the dependent population living at home.

In terms of vaccine schedule, during this 2022–2023 season, the majority of Spanish regions have chosen the simultaneous administration of flu and COVID-19 vaccines to optimize the vaccine campaign, especially for those patients who are highly vulnerable and largely dependent. In this pandemic context, it is highly relevant to encourage influenza vaccination, because recent studies have demonstrated up to 8.4% of positive flu cases in patients hospitalized with SARS-CoV-2, with increased rates of mechanical ventilation and mortality in co-infected patients [59]. This is very relevant because preventive measures are based on influenza vaccination and reduced coinfections of influenza virus and SARS-CoV-2 [60].

In recent months, various review articles describing the effectiveness of the different influenza vaccines following the GRADE methodology have been published [61,62,63,64]. This is a similar approach that has been used by international entities to evaluate these new influenza vaccines [65,66,67,68] and also by the most relevant scientific societies at the national level in the vaccination field [69]. The conclusions derived from all of these studies, comparing the different vaccines, are shown in Table 1. This table represents diverse evidence supporting the prevention of flu disease once it is confirmed by a microbiological laboratory following the criteria of the European Medicines Agency (EMA) when an efficacy/effectiveness study is considered necessary [70].

In two of the most recent publications following an evaluation by international entities, the available evidence was limited when two new vaccines were compared [64,71]. Among the four recommendations evaluated by ACIP, three showed preference for the use of a high-dose vaccine, possibly due to stronger evidence in comparison to the vaccine with the standard dose and because they included data from a randomized clinical trial that evaluated the reduction of flu disease as confirmed by a laboratory in comparison to the other influenza vaccines [72]. Finally, in August 2022, ACIP published its recommendation for adults aged > 65 years based on the available evidence and other aspects including provisions, establishing a preferential use for three vaccines (the recombinant vaccine, the high-dose vaccine or the standard-dose adjuvanted vaccine) over the non-adjuvanted classical vaccine with standard doses [73].

It is expected that in further evaluations reviewing new evidence, including novel findings, these recommendations might change. In recent months, different observational studies have evaluated the new influenza vaccines. Among them, of great interest is the study published in the context of the European Cardiology Society Meeting (ESC 2022), where the high-dose tetravalent vaccine was associated with lower hospitalization rates for pneumonia and flu in comparison to the standard tetravalent vaccine [74]. This is consistent with the findings of randomized clinical trials and previous observational studies [75].

### 3.3. Vaccination against COVID-19

Recommendations for the use of COVID-19 vaccination have been adapting to the availability of vaccines, their composition, the population vulnerability and the evolution of the virus and the pandemic. Available approved vaccines against COVID-19 have been demonstrated to be safe and effective against clinical disease, especially against severe forms, death and post-COVID-19 symptoms (“long covid”) [76]. There are several limitations with the currently available vaccines, including the inability to avoid infection and transmission, the limited duration of protection due to waning effectiveness, and the immune escape of new SARS-CoV-2 variants. All of these issues point to the need for booster doses. This scenario has also prompted the development of adapted versions of the current vaccines and continued efforts in the search for new vaccine alternatives. The recent approval by the European Medicines Agency (EMA) of new vaccines based on different technologies (VidPrevtyn Beta^®^ (Sanofi Pasteur, Lyon, France)), Valneva (Valneva Austria GmbH, Vienna, Austria) and Nuvaxovid^®^ (Novavax, Jevany, Czech Republic) and adapted versions of mRNA vaccines (Spykevax^®^ (Moderna Madrid, Spain) and Comirnaty^®^ (Pfizer, Mainz, Germany) for use as boosters offers new possibilities for protection of the population against this devastating pandemic. In this context, the NEP group has taken the following positions regarding COVID-19 vaccines:According to official Spanish recommendations, it is necessary to administer a new booster vaccine against COVID-19 (4th dose) to the most vulnerable groups (adults > 60 years old or regardless of age, all patients with high-risk factors for severe COVID-19) and to essential professional groups.The timing for administration of the booster dose should be ideally a period of at least 3 months after receiving the last vaccine dose (5 months with Comirnaty).There are other prevention interventions (non-pharmacology measures, early antiviral treatment, monoclonal antibodies), that are complementary to vaccination, and they are, indeed, recommended to many of the high-risk groups for COVID-19. These measures should be followed with the same fidelity and intensity to achieve the optimal protection coverage for the most vulnerable groups since viral evolution could also affect their effectiveness.All new and/or updated vaccines that have been approved by EMA have met all of the requirements for use in the target population following the package insert.Regarding the booster dose, there is not enough scientific evidence demonstrating the general superiority of a particular booster vaccine over the others. It is expected that the booster using bivalent vaccines would elicit a more specific and longer duration of immune response against infection by the Omicron variant. However, both booster doses using monovalent vaccines with the original strain or with multivariant vaccines (including the original strain and other variants such as beta, Omicron BA1/2 or Omicron BA4/5) are initially useful vaccines in order to reinforce the immune response against severe COVID-19 irrespective of the predominant circulating variant that is causing the disease.There are currently more than 400 vaccine candidates against COVID-19 in different clinical trials, which will increase the availability of new vaccines in the future. This aspect will be important to the evaluation of features such as the impact against transmission and/or the cross-protection activity against other SARS-CoV-2 viruses. These new generations of vaccines might modify the current vaccine strategies.The vaccine coverage using booster doses has notably decreased in comparison to the prime-boost vaccination, and in this sense, is essential to fulfill the recommended vaccination instructions against COVID-19, no matter if the patient has previously suffered the infection or not.We are probably looking towards a COVID-19 vaccination strategy based on booster doses only, aiming for specific high-risk populations. The type and composition of vaccine and interval between doses have not been established yet.

### 3.4. Burden of Disease and Future Preventive Strategies against RSV

RSV has been associated with substantial mortality among the elderly [77], being one of the main etiologic agents of lower respiratory tract diseases in adults [78]. In older adults, RSV is responsible for more primary care visits, more hospitalizations, and more deaths than the influenza virus [79]. A retrospective study that included more than 2800 adults hospitalized for RSV infection showed that most RSV deaths occurred in people aged >60 years (64%) [80]. In fact, in the United Kingdom, about 8482 adults die each year from causes attributable to RSV infections, with 93% of these deaths occurring in adults >65 years of age [79]. Mortality increases from 1 person per 100,000 in adults aged in the range of 18–49 years to 155 persons per 100,000 in adults aged >75 years [81]. Thus, while in healthy adults, RSV causes common illnesses with flu-like symptoms and frequent reinfections, in frail elderly, it causes an insidious disease that is often underdiagnosed and with high mortality [81]. Other profiles of adult patients affected by this virus are those with chronic lung diseases and immunocompromised patients, including those who have received a lung or hematopoietic stem cell transplant [82].

On the other hand, RSV infection has not only been associated with significant morbidity, but also with a large socioeconomic burden worldwide [83]. The most severe forms of this disease, represented by hospitalizations, often result in chronic respiratory morbidity after the first severe RSV infection [84]. RSV infection was previously considered an exclusively pediatric concern, although in recent years, it has been increasingly recognized as a cause of morbidity in adults [85]. The annual incidence of LRVI-RSV is 6.7/1000 in elderly patients and 37.6/1000 in patients who also have comorbidities [86]. The hospitalization and in-hospital death rates for older adults are 4.8/1000 and 1.6%, respectively, and for those with comorbidities, they are 13.2/1000 and 11.7%, respectively [86]. In a recent study comparing the clinical burden of RSV versus influenza in hospitalized elderly adults at two hospitals in Switzerland, it was observed that more RSV-infected patients had a thoracic infiltrate shown on lung radiography upon admission to hospital compared with those hospitalized for influenza virus infection [87]. Hence, more RSV-infected patients died or were admitted to ICU after 7 days of hospitalization compared with influenza-infected patients [87].

In addition, RSV disease in adults ≥50 years of age has been shown to significantly impact patients’ daily lives, including productivity, social activities, sleep, and physical function.

For all of these reasons, future treatment options and prevention measures could have a great impact on the burden of respiratory disease in adults and thus reduce the morbidity attributable to this virus [79]. In fact, the recognition of the burden associated with RSV has made vaccine development a global priority [88]. The World Health Organization (WHO) has promoted an R&D plan to facilitate the development and implementation of vaccines and monoclonal antibodies in low- and middle-income countries [89]. Encouragingly, it is estimated that immunization will be available soon [89].

Currently, 33 RSV prevention candidates are in different phases of clinical development, nine of them in phase 3 [90]. Regarding active immunization strategies, live attenuated vaccines are being developed for infants aged >6 months. Subunit vaccines are in late development phases, including maternal vaccines to protect infants, and vector, subunit and nucleic acid vaccines that would protect older adults [90]. Among the vaccine candidates for adults aged >60 years, five are currently in phase 3: RSVpreF (RENOIR Study) from Pfizer, RSVpreF/AS01 (AReSVi 006 Study) from GSK, RSVpreF/Ad26 (EVERGREEN Study) from Johnson & Johnson, MVA-BN-RSV (VANIR Study) from Bavarian Nordic, and mRNA-1345/nanoparticle (ConquerRSV Study) from Moderna [91].

Preliminary efficacy results of the phase III clinical trial of Pfizer’s bivalent RSV vaccine candidate (RSVpreF) for older adults (RENOIR study, NCT05035212), have been recently published in IDWeek. This study contains data from approximately 34,000 participants of 60 years of age and older, who were randomized (1:1) to a 120 μg dose of RSVpreF or placebo. It has been shown that RSVpreF has a vaccine efficacy of 66.7% (96.66% CI: 28.8–85.8%) for the prevention of RSV-associated lower respiratory tract infections. The vaccine efficacy was 85.7% (96.66% CI: 32.0–98.7%) in preventing the most severe disease, defined by the presence of three or more symptoms associated with RSV, and the vaccine was well tolerated, with no safety concerns. Based on these results, Pfizer intends to submit the dossier for regulatory approval in the coming months.

Preliminary efficacy results of the phase III clinical trial of GSK’s vaccine candidate (RSVPreF3 OA AS01) for older adults (AReSVi 006 Study) were recently published in IDWeek [92]. The study includes data from 25,000 participants of 60 years of age and older, who were randomized (1:1) to receive one dose of RSV PreF3 OA vaccine or placebo prior to RSV season. The candidate demonstrated an overall efficacy of 82.6% (95% CI 57.9–94.1) and efficacy of 94.1% (95% CI, 62.4–99.9, n = 1 of 12,466 vs. 17 of 12,494) against lower respiratory tract disease (RSV-LRTD). In patients with pre-existing comorbidities, such as cardiorespiratory and endocrine-metabolic pathologies as underlying diseases, vaccine efficacy was 94.6% (95% CI, 65.9–99.9, n = 1 of 4937 vs. 18 of 4861), with a 93.8% (95% CI, 60.2–99.9, 1 of 4487 vs. 16 of 4487) efficacy observed in adults 70–79 years of age [92]. Furthermore, vaccine efficacy against RSV-LRTD was consistent for the RSV-A and RSV-B subtypes.

Overall, given all of these extraordinary results and the ambitious horizon with several RSV vaccine candidates rapidly progressing, vaccination will become a real preventive option in the near future.

## 4. Discussion

The SARS-CoV-2 pandemic has changed the relative impact of the traditional CAP pathogens worldwide, making them the most frequent etiologic agent of pneumonia in adults in the current epidemiological setting. Although the circulation and prevalence of other common respiratory pathogens such as flu, RSV or pneumococci declined during the first pandemic years [93,94,95], with the return to normal habits, these pathogens are also returning, as has been confirmed in different reports [94,95,96]. These data reinforce the importance of vaccination against all possible vaccine-preventable causes of CAP, including not only COVID-19, but also influenza, VRS and pneumococcus, among others. In addition, vaccine efficacy against respiratory pathogens has been shown to be greater for aged females than for males and, therefore, sex should be considered as a variable in vaccine trials in aged populations [97].

Pneumococcal conjugate vaccines (PCVs) for adults in Spain are receiving more support as the number of Spanish regions that include them for adults is increasing; they are being used as a single dose for the immunocompetent population or in a sequential schedule with PPSV23 for high-risk patients or those with comorbidities. It would be very useful to standardize all of the different vaccine calendars in the different Spanish regions on the basis of a unique common national calendar for adults. With the arrival of the new PCV15 and PCV20 vaccines, it might be the right time to implement a common vaccine strategy for the adult population in the 19 different Spanish regions. Furthermore, this winter is the best time to start concomitantly with the seasonal influenza and COVID-19 fourth-dose campaigns.

In addition, the arrival of new influenza vaccines designed for the most vulnerable population, such as the high-dose tetravalent vaccine, will allow increased protection against the influenza virus and its derived complications. Influenza vaccination requires increased coverage rates, especially in the elderly population and risk groups of patients with comorbidities. It would be very useful to obtain publicly available vaccine coverage for the groups mentioned above to implement better policies and decrease the complications of influenza infections at the national level. The arrival of newer influenza vaccines with reinforced immunogenicity, such as the high-dose tetravalent vaccine, will increase the protection against flu and its complications among members of the most vulnerable population. The high-dose vaccine is based on scientific evidence of great relevance that is sufficient to support its preferential use in comparison to the standard influenza vaccines for this population group. Finally, the use of a systematic influenza vaccination program targeting the pediatric population for the next season that has already started in several Spanish regions (2022–2023) will contribute to the successful prevention of this disease among adults. However, more intensive efforts to increase and maintain vaccine coverage rates for these vaccines against CAP will be necessary.

In terms of COVID-19 vaccination, a second booster dose using monovalent vaccine with the original strain has been demonstrated to be immunogenic and effective against symptomatic infection, hospitalization, severe COVID-19 and death. For instance, the risk of dying during the initial Omicron period was 3.8 times lower in people who received a second booster dose compared to those vaccinated with only one booster dose [98]. However, the effectiveness of vaccination in preventing the hospitalization of adults with the second booster during the BA.4/BA.5 period (60%; 95% CI, 42–73%) was lower compared to the BA.2/BA.2.12.1 period (80%; 95% CI, 71–81%) [99]. It is expected that immunosuppressed individuals will mount an even weaker response against Omicron, as was shown after the first booster [100]. Booster doses using bivalent vaccines including the original and BA.5 variants have shown four times more neutralizing antibody levels against BA.5 and 10 times more against BQ.1.1 compared to the monovalent vaccine [101]. The major companies, Pfizer and Moderna, have communicated similar results in an exploratory analysis out of their phase 2/3 clinical trials. The important role of synthetic nanoparticles in the development of new vaccines leading to improved immunogenicity, efficacy and safety has been well-established. In the case of mRNA vaccines, they became fundamental not only as carriers but also in overcoming major challenges involving stability, cell penetration, endosomal release, a certain degree of tissue tropism, and immune potentiation [102]. Among the different nanoparticles-based platforms, a lipid–nanoparticle (LNP) complex is taking the lead, and among its main features are cholesterol, structural phospholipids, polyethylene glycol (PEG) lipids and ionizable lipids, each playing a definite role in these formulations [102].

Pertussis is also a cause of pneumonia in adults. The incidence of pertussis has been generally increasing in adults across the world since the year 2000 even though the true burden is often underestimated due to the atypical nature of the disease, challenges in diagnosis, and lack of suspicion among healthcare professionals. In the USA, the proportion of adults aged ≥65 years needing hospitalization increased significantly from 2000 to 2016 [103,104]. Certain patients with underlying conditions (asthma, chronic obstructive pulmonary disease (COPD), or obesity) are potentially at increased risk of pertussis infection. The addition of other factors including immunodeficiency and smoking seems to be associated with worsened pertussis symptoms and with an increased pertussis-related hospitalization rate. On the other hand, asthma and COPD might be exacerbated by pertussis, affecting patient quality of life and increasing health care resource utilization and direct medical costs [105,106,107]. Hence, it is important to vaccinate against this respiratory infection in adults who have not received this vaccine previously. As a global strategy to prevent respiratory infections that cause pneumonia in the adult population, this recommendation is even more relevant for high-risk patients with comorbidities.

## 5. Conclusions

Our proposed vaccine schedule against adult CAP pivots on those infections for which a specific vaccine with an indication for the adult population is available. Our recommendations do not aim to interfere with the official recommendations of any country that may vary for different reasons, from epidemiological factors to economic limitations. However, this vaccine calendar to prevent CAP in adults is focused in the individual perspective of protection and guided by the intention of assisting health care professionals in their daily practice with the best possible schedule for vaccine protection against CAP according to currently available evidence. We believe that our initiative, which has gained important acceptance within our country, can be extrapolated to others and contribute to the scientific and educational support and practical guidance of health professionals attending target populations with CAP and aiming to establish the best clinical recommendation based on the most recent scientific evidence and the individual circumstances of each patient.

## Figures and Tables

**Figure 1 antibiotics-12-00138-f001:**
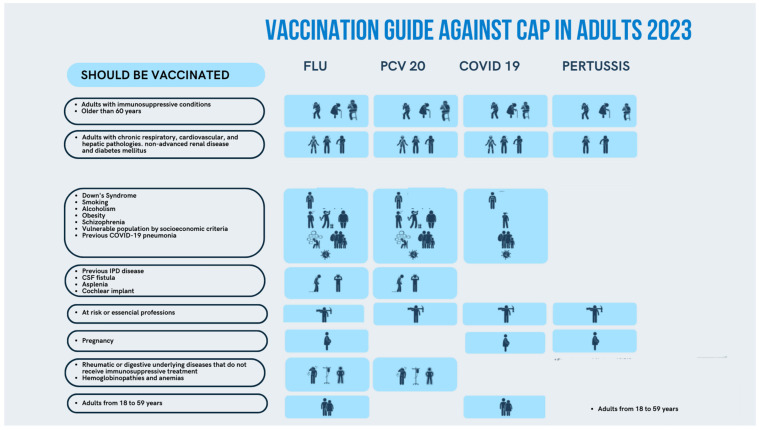
Vaccination guide against community-acquired pneumonia (CAP) in adults caused by vaccine-preventable diseases.

**Figure 2 antibiotics-12-00138-f002:**
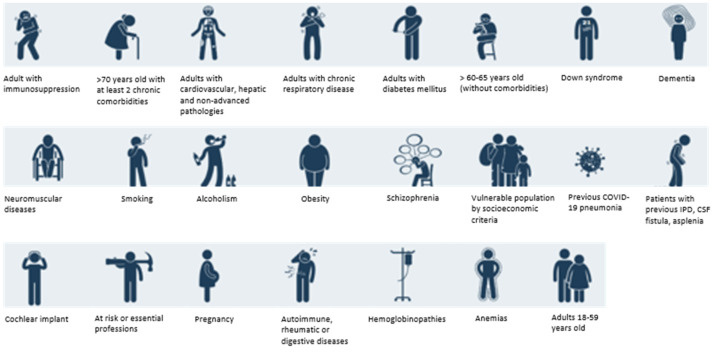
Icon interpretation from the vaccination guide.

**Figure 3 antibiotics-12-00138-f003:**
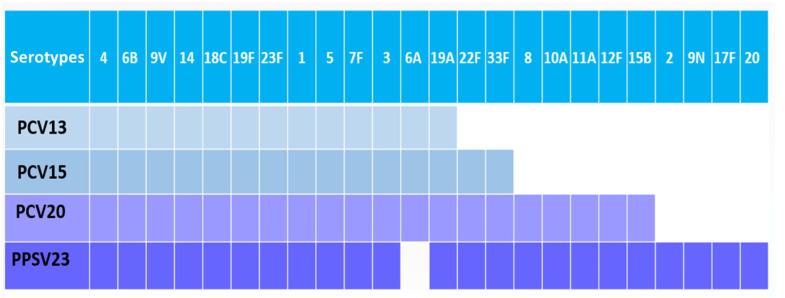
Serotypes included in the different pneumococcal conjugate vaccines (PCV13, PCV15 and PCV20) or in the 23-valent pneumococcal polysaccharide vaccine (PPSV23).

**Table 1 antibiotics-12-00138-t001:** Evaluation of the scientific evidence for new influenza vaccines in comparison to classical vaccines in the prevention of flu affecting adults, as confirmed by a microbiological laboratory. The table includes relevant evidence from different international entities such as NACI (Canada), ECDC (Europe), STIKO (Germany), ATAGI (Australia) and ACIP (USA).

Entity	NACI [66,67,71]	ECDC [65]	STIKO [68]	ATAGI [61,62,63]	ACIP [64]
Publication date	May 2018 (HD and adj) August 2020 (TC) September 2022 (REC)	October 2020	January 2021	November 2020 (adj) March 2021 (TC) March 2022 (HD)	August 2022
HD	A (Maximum)	+++ (Moderate)	++++ (High)	++ (Low) +++ (moderate)	High Support HD
Recombinant	B (Limited)	+++ (Moderate)	+++ (Moderate)	Not evaluated	Moderate Do not support one against the other
Tissue culture	I (Insufficient)	No evidence	++ (Low)	+ (very low)	Not evaluated
Adjuvanted	I (Insufficient) *	No evidence	++ (Low)	+ (very low)	Moderate * Do not support one against the other

HD (flu vaccine of high dose), Adj (flu vaccine adjuvanted), TC (flu vaccine of tissue culture), REC (flu recombinant vaccine), * (prevention of flu, but not specified that flu was confirmed by laboratory). Levels of GRADE method: ++++ (High certainty degree), +++ (Moderate certainty degree), ++ (Low certainty degree), + (very low certainty degree). NACI follows a methodology similar to to that of GRADE: A (Maximum certainty degree), B (Limited certainty degree), I (Insufficient certainty degree).

## Data Availability

Not applicable.

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
