# Peer review of "Vaccination against Community-Acquired Pneumonia in Spanish Adults: Practical Recommendations by the NeumoExperts Prevention Group"

_antibiotics, 2023, doi:10.3390/antibiotics12010138_

Round 1

Reviewer 1 Report

Redondo et al performed a review of literature on the available vaccines against the microbes responsible for community acquired pneumonia and the recommendations put forth regarding these vaccinations. The manuscript is well written, but the organization of this review article is quite confusing. 

1. Under section 2. Results, subsections included 2.1, 2.1.1, 2.2.2, and then 3.1 and even 4.1, but there are also section 3. Discussion and section 4. Materials and Methods. The subsections under section 2. Results will need to be revised and relabeled to avoid confusion.

2. Materials and methods should be moved after Introduction and before results. 

3. Numerous grammatical mistakes exist in this manuscript, and as a result, this manuscript will need to be edited extensively for grammar to improve readability. 

Author Response

“Redondo et al performed a review of literature on the available vaccines against the microbes responsible for community acquired pneumonia and the recommendations put forth regarding these vaccinations. The manuscript is well written, but the organization of this review article is quite confusing. 

  1. Under section 2. Results, subsections included 2.1, 2.1.1, 2.2.2, and then 3.1 and even 4.1, but there are also section 3. Discussion and section 4. Materials and Methods. The subsections under section 2. Results will need to be revised and relabeled to avoid confusion..

We totally agree  with the reviewer and we have modified the sections and subsections following the suggestions. We have moved the material and methods section just after the introduction and therefore the results start in section 3.

Please, see the new version submitted.

 “2. Materials and methods should be moved after Introduction and before results

We have moved them to section 2 (after introduction and before results). Please, see the new version submitted.

  1. “Numerous grammatical mistakes exist in this manuscript, and as a result, this manuscript will need to be edited extensively for grammar to improve readability”.

We have performed an extensive review of the English language within the article, as there were several grammar and typo mistakes throughout the manuscript. We apologize for all these mistakes. Please, see the manuscript with tracked changes to see all the changes in the text.

Reviewer 2 Report

the information is very consistent, especially the handling of atypical and localized investigations, however, and despite the fact that it says that the methodology described by the citations [27-29] was used, it is necessary to specify within the work the criteria that were taken in bill.

Author Response

  1. “the information is very consistent, especially the handling of atypical and localized investigations, however, and despite the fact that it says that the methodology described by the citations [27-29] was used, it is necessary to specify within the work the criteria that were taken in bill.”.

 In the new version, we have included that two virtual and face-to-face meetings were organized to discuss the positioning of our group and we have included the reference that we follow using the Oxford criteria. This cite was not included in the original version submitted and is the criteria that we took.

Please, see the new methods section.

Reviewer 3 Report

AS you have indicated IPD is emerging as an important cause of adult mortality. There are many concerns  for MDR and your paper is timely one. 

However the abstract is little superficial and does not truly reflect the content of the manuscript.

Your vaccination guide is a relevant and ready to use one .

Further validation at population level  is needed  before recommendation for day to day use and as policy 

Author Response

“As you have indicated IPD is emerging as an important cause of adult mortality. There are many concerns for MDR and your paper is timely one.

However the abstract is little superficial and does not truly reflect the content of the manuscript.

Your vaccination guide is a relevant and ready to use one.

Further validation at population level is needed before recommendation for day to day use and as policy”

We thank the Reviewer for the comments about the manuscript. We have added a new paragraph in the abstract of the new version submitted to indicate the relevance of the recommendations including the impact of vaccination against antibiotic resistance. We could not add more details because the abstract has a limited number of words. 

Reviewer 4 Report

I like the work, and it is of the highest importance. The paper is well written, but I have some suggestions; maybe those will help to improve the soundness and novelty of this paper.

1-The outputs of the review study need to be highlighted in an abstract.

2-In the introduction section, a brief description of new insights into disease prevention, such as Q fever and the role of viruses and Mycoplasma pneumonia in Spain, is required.

3-Is there any significant research on the impact of vaccination on males or females?

4-The use of nanotechnology to boost COVID-19's immune system that could be added. 

Author Response

“I like the work, and it is of the highest importance. The paper is well written, but I have some suggestions; maybe those will help to improve the soundness and novelty of this paper.”

1-“The outputs of the review study need to be highlighted in an abstract”t.

We thank the Reviewer for the comments about the manuscript. The critic about the abstract was also raised by Reviewer 3. In the new version submitted, we have added an additional paragraph explaining the relevance of the recommendations including the impact of vaccination against antibiotic resistance. We could not add more details because the abstract has a limited number of words. 

 2-“In the introduction section, a brief description of new insights into disease prevention, such as Q fever and the role of viruses and Mycoplasma pneumonia in Spain, is required”

The manuscript is based in practical recommendations against the most common vaccine-preventable diseases producing community-acquired pneumonia that includes S. pneumoniae, SARS-CoV-2, Influenza, Respiratory Syncytial Virus and Pertussis.

Following the suggestion made by the Reviewer, we have included a new paragraph in the introduction section explaining that the estimated prevalence of atypical pathogens in hospitalized CAP patients is low during a non-epidemic season. Please, see lines 89-90 of the new version submitted.

  3-“Is there any significant research on the impact of vaccination on males or females?”

In the Discussion section of the new version submitted (lines 583-585) we have added a new paragraph indicating that vaccine efficacy against respiratory pathogens has been shown to be greater for aged females than for males and therefore, sex should be considered as a variable in vaccine trials in aged populations. In addition, we have added a new reference for this cite (reference by Fink AL et al, doi:10.1152/physiol.00035.2015).

4-“The use of nanotechnology to boost COVID-19's immune system that could be added”

Following the Reviewer’s suggestion, we have explained the use of nanotechnology in the development of vaccines against COVID-19. Please see the Discussion section of the new version submitted (Lines 628-635) and the new reference by Guerrini C et al, published in Nature Nanotechnology (doi:10.1038/s41565-022-01129-w).